# A Modeling Approach for Analyzing the Hydrological Impacts of the Agribusiness Land-Use Scenarios in an Amazon Basin

**Zandra A. Cunha** [1], **Carlos R. Mello** [2,3,*], **Samuel Beskow** [1], **Marcelle M. Vargas** [1], **Jorge A. Guzman** [3] and **Maíra M. Moura** [1]

[1] Water Resources Graduate Program, Federal University of Pelotas de Pelotas (UFPel), Gomes Carneito Street 1, Porto, Pelotas 96010-610, RS, Brazil; zandra.cunha@ufpel.edu.br (Z.A.C.); samuel.beskow@ufpel.edu.br (S.B.); marcelle.vargas@ufpel.edu.br (M.M.V.); maira.moura@ufpel.edu.br (M.M.M.)

[2] Water Resources Department, School of Engineering, Federal University of Lavras (UFLA), Lavras 37200-900, MG, Brazil

[3] Department of Agricultural and Biological Engineering, College of ACES, University of Illinois at Ubana-Champaign, Urbana, IL 61801, USA; jag@illinois.edu

* Correspondence: crmello@ufla.br

**Abstract:** The Xingu River Basin (XRB) in the Brazilian Amazon region has a great relevance to the development of northern Brazil because of the Belo Monte hydropower plant and its crescent agribusiness expansion. This study aimed to evaluate the potential of the Lavras Simulation of the Hydrology (LASH) model to represent the main hydrological processes in the XRB and simulate the hydrological impacts in the face of land-use change scenarios. Following the trend of the most relevant agribusiness evolution in the XRB, four agribusiness scenarios (S) were structured considering the increase in grasslands ($S_1$: 50% over the native forest; $S_2$: 100% over the native forest) and soybean plantations ($S_3$: 50% over the native forest; $S_4$: 100% over native forest). Average hydrographs were simulated, and the frequency duration curves (FDC) and average annual values of the main hydrological components for each scenario were compared. The results showed that, in general, changes in land use based on deforestation in the XRB would lead to an increase in flood streamflow and a reduction in baseflow. The increases in direct surface runoff varied from 4.4% for $S_1$ to 29.8% for $S_4$ scenarios. The reduction in baseflow varied from −1.6% for $S_1$ to −4.9% for $S_2$. These changes were reduced when the entire XRB was analyzed, but notable for the sub-basins in its headwater region, where the scenarios were more effective.

**Keywords:** Amazon region; Xingu River Basin; land-use changes; hydrological impacts





## 1. Introduction

Agribusiness has been the primary economic activity in Brazil in recent decades. Soybean and beef are Brazil's significant expansion of this agribusiness, which has changed its land use, mainly towards the Amazon and Cerrado biomes, because these commodities are destined for the international market. These transactions grant expressive economic resources to the country. However, this expansion has threatened its ecology and hydrological functions, which need to be studied or addressed, especially in the Amazon basin [1,2].

The Xingu River Basin drains an area of approximately 530,000 km², mainly in the Brazilian Amazon region. This basin has a great relevance to the development of northern Brazil, mainly because of the Belo Monte hydropower plant and its crescent agribusiness expansion.

The Brazilian Cerrado and Amazon Forest biomes (and the transition between them) are found in the Xingu River Basin (XRB). However, agribusiness activities have steadily increased in the XRB's headwaters, which has led to continuous deforestation [3]. The conversion from tropical forests into grasslands for livestock (beef production) and annual

crops, mainly soybean, are the leading causes of deforestation in both the Amazon Forest and Brazilian Savanna (Cerrado) [4].

Land-use and soil cover changes are directly linked to the basins' hydrology [5]. These alterations primarily affect the basins' evapotranspiration, ultimately changing its streamflow regime and soil water storage. Hydrological models allow for an understanding of these impacts and their well-known limitations in simulating the complex interactions between soil and plants [4,5].

Some studies have been developed for the Amazon Forest biome, including the XRB, considering land-use scenarios. Changes in the water balance in the Xingu Basin, regarding the variations in the climate and deforestation between 1970 and 2000, based on the Integrated Biosphere Simulator (IBIS) hydrological model, were evaluated by [3]. The study by [6] investigated the potential impacts caused by grasslands instead of native forests to the streamflow and other water balance components in the Iriri basin, an affluent of the XRB, through the Soil Water Assessment Tool (SWAT) model. Hydrological alterations, taking climate change and land-use/soil cover scenarios by means of the Ecosystem Demography model, version 2, (ED2+R) hydraulic river model in the Tapajós River Basin, were assessed by [7]. Using the "Modelo para Grandes Bacias—Instituto de Pesquisas Hidraulicas" (MGB-IPH) model, [8] assessed the potential hydrological impacts caused by increased grasslands and agriculture over the Amazon Forest in the Mortes River watershed in the Araguaia basin.

Hydrological models have been developed to reduce the uncertainties related to hydrological processes. However, they are complex when being calibrated in tropical regions because of the number of soil- and plant-related parameters, which are difficult to obtain in the literature or laboratory [5]. The Lavras Simulation of Hydrology (LASH) was developed based on the datasets available in Brazil, requiring fewer parameters than most models incorporating the accumulated experience with the tropical soil–plant–atmosphere relationships [9]. When calibrating the LASH, we explored simulating the extreme streamflows and water balance adequately and confirmed if the calibrated parameters followed the physical reality of the basin concerning its hydrological processes. These parameters comprised intervals encompassing the observations made under field conditions during the numerical calibration [9–11].

The LASH model was developed by [9–11] and, more recently, improved by [12]. In Brazil, this model has been used to simulate the hydrological behavior in basins in the southeast and south Brazilian regions, including climate change and land-use/soil cover scenarios [13–15]. In all these applications, the LASH generated an excellent accuracy for the streamflow and could capture these basins' hydrological patterns, including simulations of the spatial and temporal soil moisture [8].

This study brings some scientific contributions to hydrology in the tropics, mainly for the Amazon region, presenting the LASH model's applicability to a basin with unique land-use/soil cover characteristics, i.e., a combination of the Amazon Forest, Brazilian Savanna, and a transition between these biomes. Because of the model's capability of simulating land-use scenarios with an acceptable performance and lower level of uncertainty [13,14], we sought to simulate hydrological impacts from agribusiness scenarios, focused on soybean and grasslands instead of the native forest in the Xingu River Basin, Brazilian Amazon. For that, the LASH model was calibrated and validated using the current land uses, and then it was run over the scenarios, changing the vegetation parameters used by this model. The scenarios were designed based on the trend in the land use/land cover in the basin in recent decades, on the deforestation for soybean and grasslands (agriculture and livestock), and considered changes of 50% and 100% in the sub-basins with the strongest trends.

## 2. Materials and Methods

### 2.1. The Xingu River Basin (XRB)

The XRB is one of the main tributaries of the Amazon River Basin (Figure 1). It comprises a drainage area of approximately 530,000 km$^2$, covering the states of Mato

Grosso and Pará. From its springs, located in Mato Grosso, to its outlet, the Xingu River has a length of 1640 km [3]. The XRB is necessary for Brazil's economic and environmental context, mainly for its hydraulic potential, biodiversity, and ecosystem services [3].

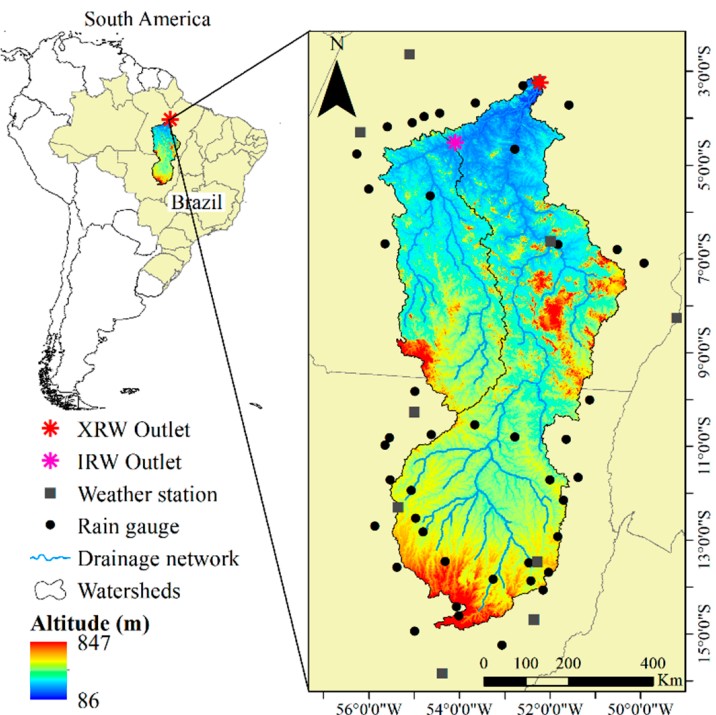

**Figure 1.** Location of the Xingu River basin and the digital elevation model, rain-gauge, and weather stations, and fluviometric stations used in this study.

The drainage area considered in this study was determined on the Altamira fluviometric station (code 18850000, Agência Nacional de Águas—ANA), since this is the last gauge before the Belo Monte hydropower plant with a satisfactory streamflow historical series for hydrological studies (Figure 1). Thus, the XRB's area in this study accounts for 448,022.80 km$^2$. Approximately 92.3% of the basin flows through the Amazon biome and 7.7% through the Brazilian Savanna (Cerrado).

According to the Köppen-type climate [16], the XRB comprises three types (Figure 2a): Af (humid or super humid tropical, without a dry season), Am (monsoon climate), and Aw (tropical with a dry winter). The Af climate is characterized by an annual precipitation depth between 2200 and 2700 mm and a mean annual temperature above 26 °C [16]. The Am climate represents the largest area of the basin, under a rainfall with a north–south gradient, covering a region from the southern Pará State to the north of the Mato Grosso State [16]. In southwestern Mato Grosso, the annual precipitation decreases to approximately 2000 mm, and a dry winter characterizes the Aw climate [16]. The dry season in the XRB occurs from May to October, and the rainy season is between November and April [3].

Based on the Soils Map of the Legal Amazon Forest published by the Geosciences Directorate of the Brazilian Institute of Geography and Statistics (IBGE) in 2012, with a scale of 1:250,000 [17] (Figure 2b), the predominant soil classes in the XRB are Ultisols (Argisols) (53.1%) and Oxisols (26.4%). Inceptisols (2.7%), Histosols (3.2%), Entisols (Litholic Neosol) (10.4%), Ultisols (Nitosol) (2.3%), Histosols (Organosol) (0.0006%), and Plintic (Plintosol) (1.8%) classes are also present.

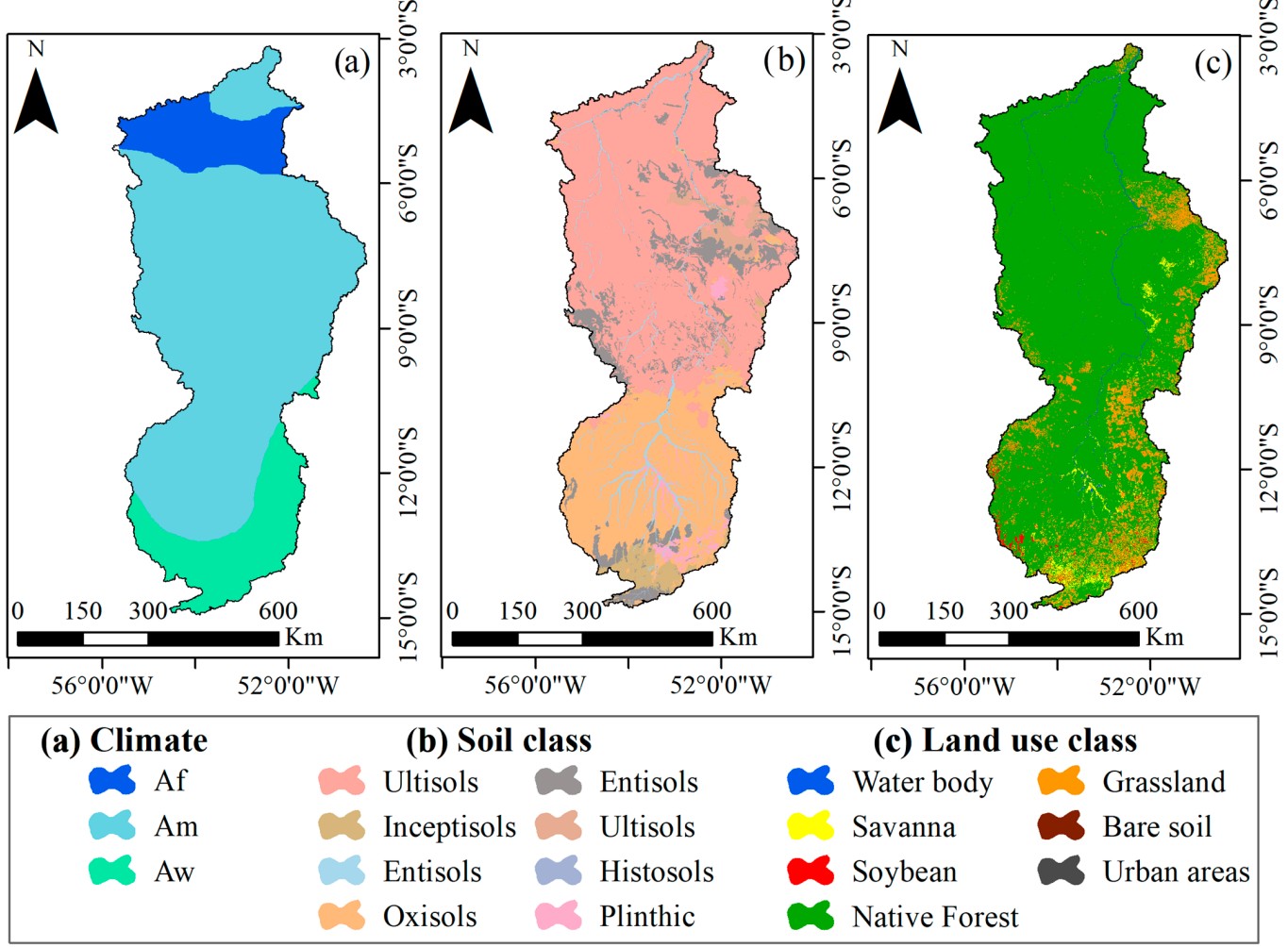

**Figure 2.** Köppen-type climate (**a**), soil classes (**b**), and land-use classes (**c**) in the XRB.

According to the Annual Mapping of Land Cover and Land Use in Brazil [18] for 2000, the predominant land use class in the XRB is native forest, covering 86.5% of the basin's area. It is essential to highlight that the expansion of agribusiness in the XRB occurred mainly from the year 2000 due to an increase in the values of commodities worldwide. The remainder comprises soybean (0.6%), grasslands (9.6%), bare soil (0.08%), urbanization (0.007%), and water bodies (0.8%) (Figure 2c).

### 2.2. The LASH Model Structure

The LASH model simulates the behavior of the hydrological cycle in a watershed, estimating, in different time steps, the streamflow, soil–water storage, and evapotranspiration [11]. It is a hybrid distributed model that combines conceptual and physical principles for long-term hydrological simulations [9]. In this study, we used a version of the LASH model that uses a semi-distributed spatial discretization (sub-basins) with daily simulations and the MATLAB® programming language [12]. Briefly, the runoff structure of the LASH is based on three modules: (i) direct surface runoff, based on the modified CN method [19]; (ii) subsurface runoff, and (iii) baseflow, both based on the Brooks and Corey equation for the hydraulic properties of porous media [20].

After quantifying these runoff components, the LASH model converts each component into outflow by determining three linear reservoirs in each sub-basin. At each time step adopted, the estimated discharge in each sub-basin is given by the sum of the flows from

the surface, subsurface, and underground reservoirs. The streamflow-routing module in this LASH version consists of the Kinematic Wave method [21,22].

The evapotranspiration in the LASH model can be summarized as follows. Rainfall is stored in the canopy up to its saturation. The model estimates the canopy storage capacity (CSC) for each sub-basin in the spatial discretization setup.

$$\mathrm{CSR} = \alpha \cdot \mathrm{LAI} \tag{1}$$

CSR is the interception storage reservoir (mm), $\alpha$ is the maximum storage capacity per canopy area, and LAI is the leaf area index ($m^2 m^{-2}$). After the CSR determination for the sub-basins, the LASH model runs the water balance in the interception reservoir, taking the storage depth in time $t - 1$ ($I_{t-1}$), total rainfall in time t (Rt), and canopy storage in time t ($I_t$), considering:

If $I_{t-1} = 0$ and $R_t >$ CSR or $R_{t-1} >$ CSR, then $I_t =$ CSR

If $I_{t-1} = 0$ and $R_t <$ CSR, then $I_t = R_t$

If $I_{t-1} + R_t <$ CSR, then $I_t = I_{t-1}$

After these considerations, the model estimates the rainfall (R) that hits the surface in time t:

$$\mathrm{R} = \mathrm{R_t} - (\mathrm{I_t} - \mathrm{I_{t-1}}) \tag{2}$$

At the end of time t, the LASH calculates the interception of the canopy in each of the sub-basins:

$$\mathrm{I_{esv}} = \mathrm{I_t} - \mathrm{ET_P} \tag{3}$$

$I_{esv}$ is the canopy interception (mm) and $ET_p$ is the potential evapotranspiration (mm).

In the LASH model, the actual evapotranspiration is calculated using the Penman–Monteith equation [23], using the physiological parameters of the vegetation.

$$\mathrm{ET_a} = \frac{0.408 \cdot \Delta \cdot (\mathrm{R_n} - \mathrm{G}) + \left( \frac{86400 \cdot \gamma \cdot \varepsilon}{\mathrm{T_{Kv}} \cdot \mathrm{R} \cdot \mathrm{r_a}} \right) \cdot (\mathrm{e_s} - \mathrm{e_a})}{\Delta + \gamma \cdot \left( 1 + \frac{\mathrm{r_s}}{\mathrm{r_a}} \right)} \tag{4}$$

$ET_a$ is the actual evapotranspiration (mm), $\Delta$ is the slope of the saturation vapor pressure ($kPa \cdot °C^{-1}$) at air temperature T (°C), Rn is the net radiation ($MJ\ m^{-2}\ d^{-1}$), $e_a$ is the actual water vapor pressure (KPa), $e_s$ is the saturation water vapor (Kpa), G is the energy flux from the soil ($MJ\ m^{-2}\ d^{-1}$), $\gamma$ is the psychrometric constant (($kPa \cdot °C^{-1}$), $\varepsilon$ is the ratio molecular weight water vapor/dry air (0.622), $T_{Kv}$ is the virtual temperature (K), R is the specific gas constant ($0.287\ KJ \cdot kg^{-1} \cdot K^{-1}$), $r_a$ is the aerodynamic resistance ($s \cdot m^{-1}$), and $r_s$ is the stomatal resistance ($s \cdot m^{-1}$).

The $ET_a/ET_p$ ratio is given by:

$$\mathrm{ET_a} = \mathrm{ET_p} \cdot \mathrm{K_S} \tag{5}$$

$K_s$ is the dimensional coefficient representing the exponential decay of this relationship from a threshold soil water storage ($A_L$). $K_s$ can be estimated considering the following relationships.

$$\mathrm{K_S} = \frac{\mathrm{Ln}(\mathrm{A_t} - \mathrm{A_{PWP}})}{\mathrm{Ln}(\mathrm{A_L} - \mathrm{A_{PWP}})} \ \text{if } \mathrm{A_t} < \mathrm{A_L} \tag{6}$$

$$\mathrm{K_S} = 1 \quad \text{if } \mathrm{A_t} \geq \mathrm{A_L} \tag{7}$$

$A_L$ is the threshold soil–water storage (mm) and $A_{PWP}$ is the soil–water storage at the permanent wilting point (mm).

### 2.3. Database for Running LASH in the XRB

The spatial distribution of the climate, precipitation, and fluviometric gauge stations is presented in Figure 1. The historical series corresponding to the streamflow and precipitation were acquired from "Agência Nacional de Águas e Saneamento Básico" (ANA) (HidroWeb portal—Sistema de Informações Hidrológicas) at the Altamira station (ANA code: 18901080). The streamflow data were determined after two levels of consistency, with the water level being collected twice a day and averaged to daily discharge. Forty-three rain gauges, with daily records, were selected according to the area of influence in the basin. Similarly, ten conventional meteorological stations were selected from the "Banco de Dados Meteorológico para Ensino e Pesquisa" (BDMEP) of the "Instituto Nacional de Meteorologia" (INMET). The datasets of the minimum and maximum temperatures, insolation, relative humidity, and wind speed were collected three times a day, according to the World Meteorological Organization pattern. Then, they were averaged by a daily value.

The datasets used in the LASH calibration and validation had their stationarity tested by the Mann-Kendal test [23,24]. We established a limit of 31 days for missing data in the annual streamflow series, and gaps in the meteorological data were filled using regressions between the rain gauges. Data quality control was conducted using the System of Hydrological Data Acquisition and Analysis (SHYDA) [25].

To characterize the topography of the basin, the Shuttle Radar Topography Mission (SRTM), with a 90 m spatial resolution, was obtained from the United States Geological Survey (USGS) to generate a Digital Elevation Model (DEM) of the basin (Figure 1). The LASH model requires spatial information on soil depth (Z), saturation soil moisture ($\theta$s), and permanent wilting soil moisture ($\theta$pmp) [9]. For the land-use classes, it is necessary to define the leaf area index (LAI), plant height (h), stomatal resistance (SR), and root system depth (Pr) [9]. The physical attributes of the soils and vegetation parameters were based on the literature and are presented in Table 1.

**Table 1.** Values and respective intervals of the albedo, values of height (h), leaf area index (LAI), stomatal resistance (SR), and root system depth (Pr) of each land-use class, and respective literature.

| Land Use Class | Albedo | h (m) | LAI ($m^2.m^{-2}$) | SR ($s.m^{-1}$) | Pr (mm) |
|---|---|---|---|---|---|
| Native forest (Amazon) | 0.13–0.18 [24] | 10 [14] | 6.25 [24] | 140 [25] | 2000 [9] |
| Undergrowth | 0.2–0.25 [25] | 0.5 [26] | 0.5 [11] | 65 [26] | 500 [23] |
| Grassland | 0.2–0.26 [25] | 0.5 [26] | 1.86–3.99 [27] | 60–80 [28] | 500 [23] |
| Soybean | 0.15–0.26 [29] | 0.0–1.1 [26] | 0.4–7.0 [30] | 60–90 [27] | 500 [23] |
| Bare soil | 0.1–0.35 [25] | 0 | 0 | 545.3 [13] | 500 [23] |
| Urbanization | 0.1–0.35 [25] | 0 | 0 | 545.3 [13] | 500 [23] |
| Waterbody | 0.12 [15] | 0 | 0 | 0 | 0 |

In addition, the following datasets are also required for the streamflow-routing module: (i) the length of the channels, (ii) the channels' slope, (iii) the channels' width, and (iv) which sub-basins drain into each watercourse. These data were computed in ArcGIS 10.5 GIS environment [29]. Furthermore, the GIS functionalities of the Hydrological Engineering Center–Geo Hydrological Model Simulation (HEC-GeoHMS) [31] were used to calculate the parameters for routing in the XRB drainage network.

### 2.4. Calibration and Validation of the LASH Model

Streamflow and meteorological data from 1995 to 2005 were used to warm up, calibrate, and validate the LASH model. The model warm-up allowed us to overcome the uncertainties associated with the initial hydrological conditions of the basin and was conducted using datasets from 1995. The calibration and validation of the model were performed considering datasets, respectively, from 1996 to 2000 and 2001 to 2005.

The automatic calibration of the parameters was performed using the Algorithm Genetically Adaptive Multiobjective (AMALGAM) [32], as described in [12]. The objective functions used were the Nash Coefficient—NS [33], its logarithmic version (NSlog), and the Pbias coefficient (%) [10]. The number of evaluations (defined as 5000) was the stopping criterion for the algorithm. With the acquisition of the 5000 calibrated parameter sets, the set that best fit, combining the three precision statistics simultaneously used for the calibration (NS, NSlog, and PBIAS), was chosen, while maintaining parsimony. The calibrated parameters of the LASH model were (i) the initial rainfall abstraction coefficient; (ii) the subsurface reservoir hydraulic conductivity (mm $d^{-1}$); (iii) the subsurface reservoir hydraulic conductivity (mm $d^{-1}$); (iv) the maximum flow density for capillary rise return (mm $d^{-1}$); (v) the surface reservoir response time (dimensionless); (vi) the subsurface reservoir response time (dimensionless); (vii) the baseflow delay time parameter (days); and (viii) Manning's coefficient (s $m^{-1/3}$) [9,10].

The hydrological modeling of the water balance components was computed following the discretization totaling 91 sub-basins, while the calibration of the parameters of the LASH model was concentrated. Furthermore, the Proxy basin test was also applied to validate the application of the model for stationary processes in upstream sub-basins [10]. In other words, sub-basins other than those used for the calibration of the model were considered in the Proxy basin test, which is essential to demonstrate the model's ability to simulate the hydrological processes in the basin [10,11]. In this case, the LASH was also validated for a sub-basin of the XRB called the Iriri River Basin (IRB) (Figure 1).

*2.5. Agribusiness Land-Use Scenarios for the XRB*

The land-use/land cover change scenarios were determined following the trend of the most relevant agribusiness area evolution in the XRB. Based on land-use analyses in recent decades, we identified the areas and respective sub-basins where these changes occurred. Most of the changes have occurred in the headwater region of the basin, which has been constantly modified by increasing deforestation to crop soybean (agriculture) and grasslands (livestock—beef production) [5]. In this context, four agribusiness scenarios were structured as follows:

- Scenario 1 (S1): 50% increase in grasslands over native forest, focusing on the regions where this practice has occurred more expressively.
- Scenario 2 (S2): 100% increase in grasslands over native forest, where this practice has been occurring more expressively.
- Scenario 3 (S3): 50% increase in soybean plantations over native forest, where this practice has been occurring more expressively.
- Scenario 4 (S4): 100% advance in soybean plantations over native forest, where this practice has been occurring more expressively.

The base scenario (S$_0$) for evaluating the impacts was the distribution of land uses observed in 2000, which was used to calibrate the LASH model. The scenarios of changes were built using ArcGIS 10.5 with the "Generate Random Points" tool [34]. This tool allowed us to choose a set of points in the land-use classes of interest for the changes. Subsequently, buffers were delimited around each point to encompass the areas impacted by the changes. S$_0$ and the scenarios are spatially depicted in Figure 3 and Table 2 shows the areas affected by each scenario.

The hydrological responses of the XRB to land-use changes were estimated for each scenario by making alterations to the LASH's vegetation index (Table 1). It is important to emphasize that, before the simulation of the scenarios, a sensitivity analysis of each vegetation index was performed to infer the magnitude of the impact provoked by each parameter separately. For the sensitivity analysis, the estimated values of the annual average flow and total annual evapotranspiration were evaluated according to the changes, considering increases and decreases of 20%, 40%, and 60% of the values initially used in each vegetation index.

The monthly average hydrographs simulated by the LASH, considering the land-use scenarios, were compared to those derived from the LASH after its calibration, i.e., $S_0$, when the agribusiness in the XRB had a significant increase. In addition, the frequency duration curve (FDC) and average annual values of the main hydrological components were analyzed for each scenario as follows: surface runoff ($D_{sup}$), baseflow ($D_b$), actual evapotranspiration (ETa), interception (It), and soil–water storage (At).

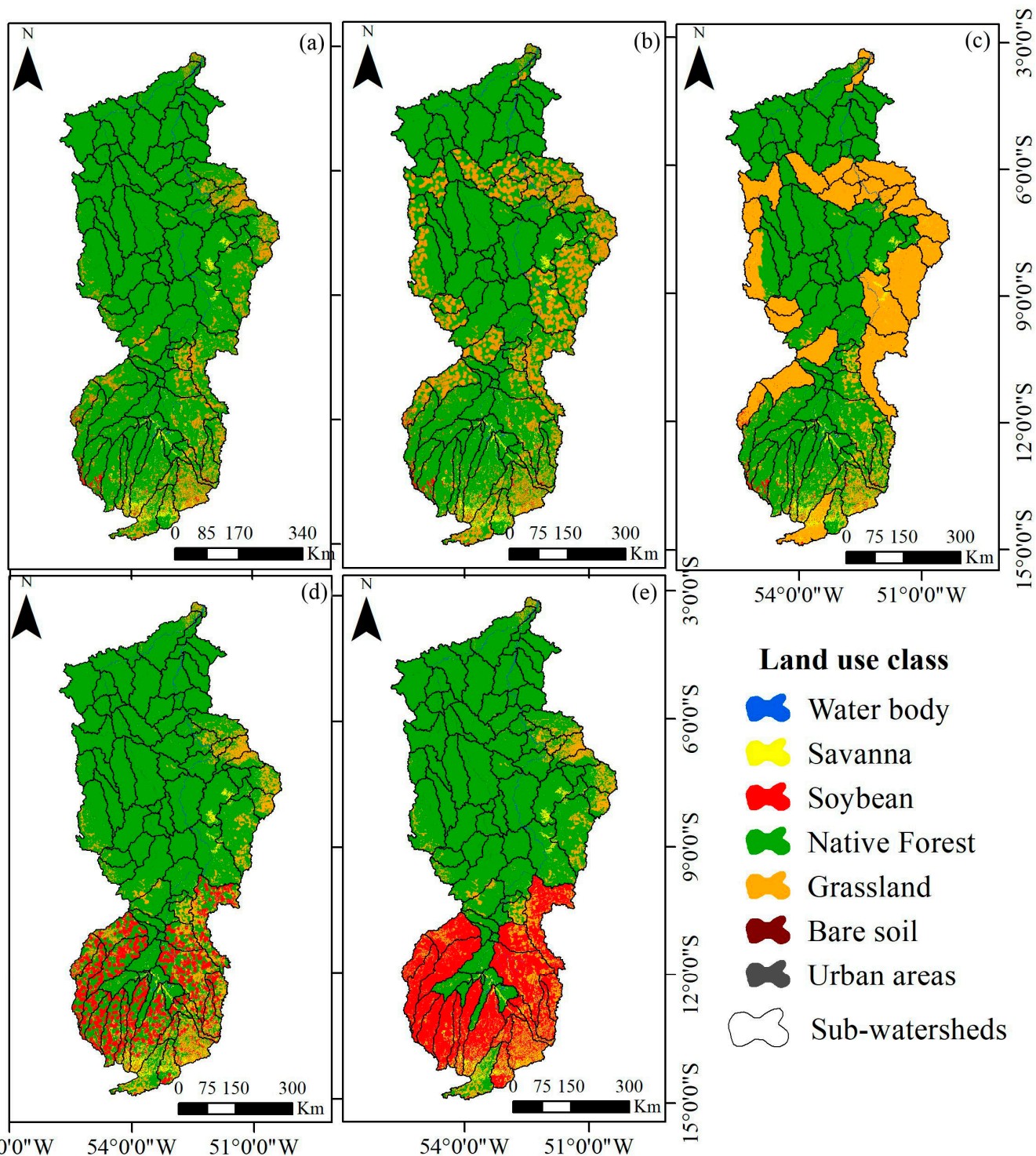

**Figure 3.** Spatial distribution of the agribusiness scenarios in XRB ((**a**) $S_0$; (**b**) $S_1$; (**c**) $S_2$; (**d**) $S_3$; and (**e**) $S_4$).

**Table 2.** Areas for the agribusiness land-use scenarios evaluated in the XRB ($S_0$: base scenario; $S_1$ and $S_2$: grassland scenarios; $S_3$ and $S_4$: soybean scenarios).

| Classes | $S_0$ (km$^2$) | $S_1$ (km$^2$) | $S_2$ (km$^2$) | $S_3$ (km$^2$) | $S_4$ (km$^2$) |
|---|---|---|---|---|---|
| Native forest (Amazon) | 387,440.6 | 341,989.3 | 277,497.4 | 344,938.9 | 294,312.6 |
| Undergrowth | 11,231.1 | 11,231.1 | 11,231.1 | 11,231.1 | 11,231.1 |
| Grassland | 2670.4 | 2670.4 | 2670.4 | 45,172.1 | 95,798.3 |
| Soybean | 42,856.6 | 88,307.9 | 152,799.8 | 42,856.6 | 42,856.6 |
| Bare soil | 357.1 | 357.1 | 357.1 | 357.1 | 357.1 |
| Urbanization | 31.8 | 31.8 | 31.8 | 31.8 | 31.8 |
| Waterbody | 3434.5 | 3434.5 | 3434.5 | 3434.5 | 3434.5 |

Further analyses of the changes were carried out considering two sub-basins of the XRB. To assess the impacts of the replacement of native forests by grasslands, the hydrological components and FDC of sub-basin 44 in the Amazon Forest biome were computed. In contrast, the hydrological components and FDC of sub-basin 80 were analyzed to assess the impacts of converting native forests into soybean plantations in the headwaters of the XRB, where the Cerrado biome is dominant.

## 3. Results and Discussion

### 3.1. Calibration and Validation of the LASH Model

Figure 4 depicts the observed and estimated mean daily hydrographs simulated for the XRB and IRB in the proxy basin test validation, the FDC for the XRB, and the calibration and validation performance statistics. The NS and NSlog statistics for the calibration and validation classify the LASH model as "very good". The values of Pbias also indicated a "very good" model fit in the calibration, while in the validation, it was "satisfactory".

Upon analyzing the average annual components of the simulated water balance in the XRB, the $ET_a$ represented the most significant component, corresponding to 53.6% of the average yearly precipitation. The simulated runoff was equal to 474.3 mm y$^{-1}$, close to the observed value (495.6 mm y$^{-1}$). According to the average estimated values of the $D_{sup}$ and $D_b$, it was observed that the latter corresponded to 41.4% of the total runoff. The hydrological and hydrodynamic of the Amazon Basin was modeled by [35], using the MGB-IPH model. They demonstrated that the surface runoff accounts for 56% of the total runoff, while the baseflow accounts for 35%. Therefore, the baseflow significantly contributes to the runoff in the XRB and could be one of the hydrological elements most impacted by the agribusiness scenarios if the soil infiltration was reduced, compromising the water yield during the dry season in the sub-basins more affected by land-use changes.

The FDC derived from the LASH was consistent with the observed FDC. However, it is possible to observe that the greatest estimated discharges were distant from those observed, with underestimations occurring. An average error of −12.6% was obtained considering all the streamflows, with an exceedance below 20%. Some difficulties in estimating the peak flows by the hydrological models were highlighted, being attributed to the inadequate representation of the spatial variability in the daily rainfall [9,10,36]. However, problems with the extrapolation of the stage-discharge rating curves cannot be ruled out, since there were significant difficulties in generating the peak flows in the daily step for large-scale basins [37]. However, for discharges with an exceedance above 80% (minimum discharges), the average error was −2.9%, meaning that there were better estimates for the baseflow.

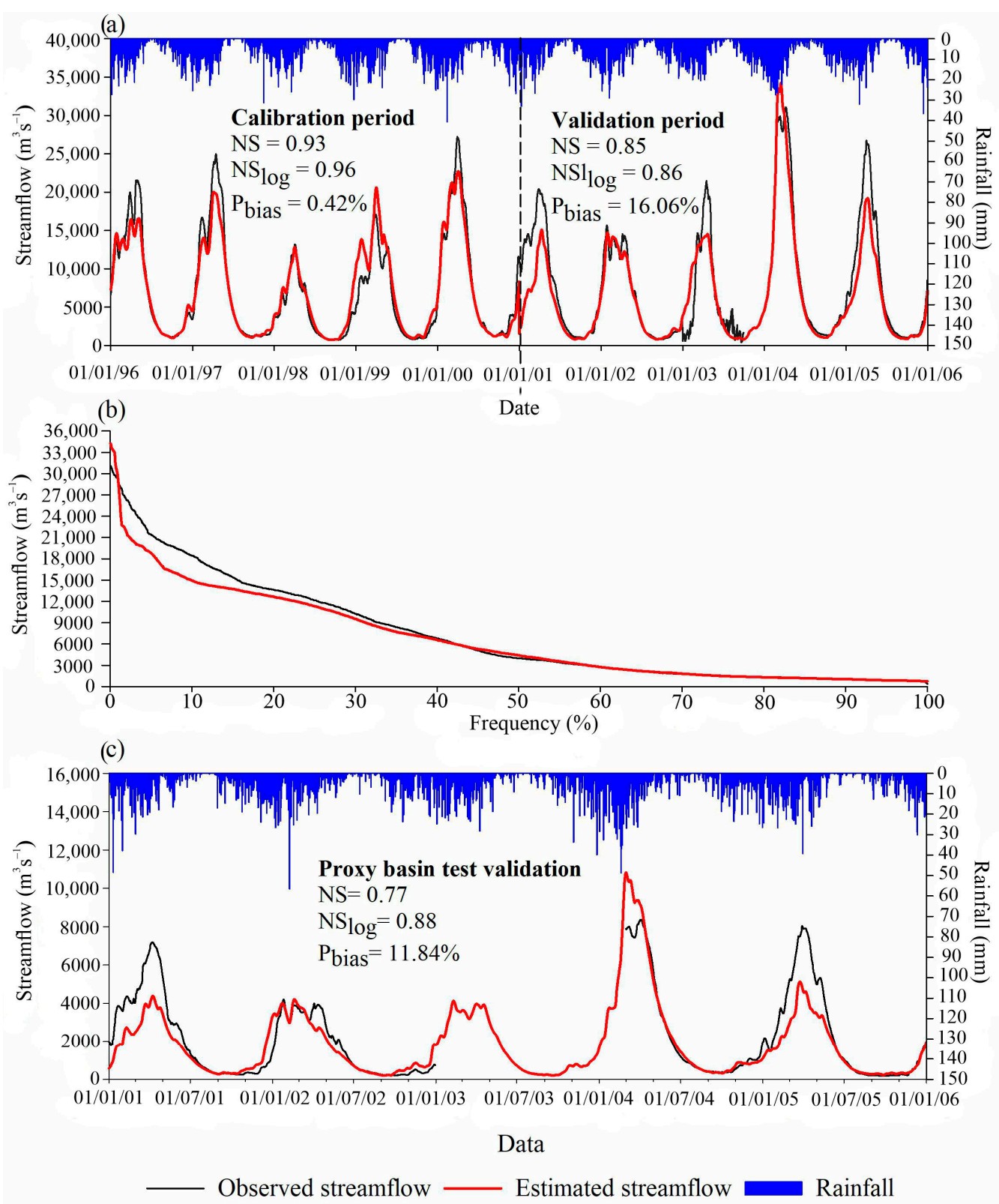

**Figure 4.** Observed and estimated daily hydrographs and respective hyetographs for calibration and validation of LASH model (**a**), observed and estimated FDC for XRB (**b**), and observed and estimated hydrographs for IRB (Proxy Basin Test) (**c**).

The LASH has been successfully evaluated in some studies with regard to its accuracy in estimating the FDC under the climate, soils, and topography of southeast and south

Brazil [11,14,15]. Analyzing the FDC estimated for the upper Grande River Basin, Minas Gerais State [36] concluded that the LASH performed well. However, the model generated a slight discrepancy in the minimum streamflows. The LASH performed well in estimating the FDC for the Jaguara Creek watershed in southern Minas Gerais, with a slight over-estimation of the Q90% streamflow [14]. Similarly, for the Fragata River watershed, in the south of Rio Grande do Sul, [15] and [11] found suitable fits of the FDC with a slight overestimation of the minimum streamflows. Therefore, the simulations of the baseflow by the LASH model were more accurate than the simulations of the peak flows, which allows for more reliable comparisons of the land-use scenarios in terms of the water yield.

The LASH validation for the IRB resulted in accurate statistics (NS and NSlog) that classified the model's performance as "very good". The value obtained for the Pbias indicated a small underestimation of the streamflows. This behavior can also be observed in the estimated hydrographs presented in Figure 4c, where most of the peak streamflows were underestimated. However, this type of validation is relevant for the model performance assessment. It transfers the model parameters obtained from the calibration using the observed hydrograph at the XRB outlet to the IRB (upstream). After defining the set of parameters for the IRB, the LASH was used to estimate the hydrograph at its outlet, which was then compared to the observed hydrograph. Thus, the model was tested using an unknown database, demonstrating adequate modeling of the hydrological processes in upstream sub-basins.

### 3.2. Sensitivity Analysis of the Vegetation Parameters in the LASH Model for the XRB

With the development and application of hydrological models to simulate the hydrological behavior of basins due to land-use changes, it is necessary to examine the models' performances at the hydrology and vegetation interfaces, considering the vegetation parameters' variation. The results obtained in the sensitivity analysis of the vegetation-related attributes in the LASH model are illustrated in Figures 5 and 6.

The albedo and vegetation height parameter variation generally did not result in relevant hydrological impacts in the XRB. Both vegetative variables caused changes in the streamflow and evapotranspiration of approximately 3% (Figure 6b).

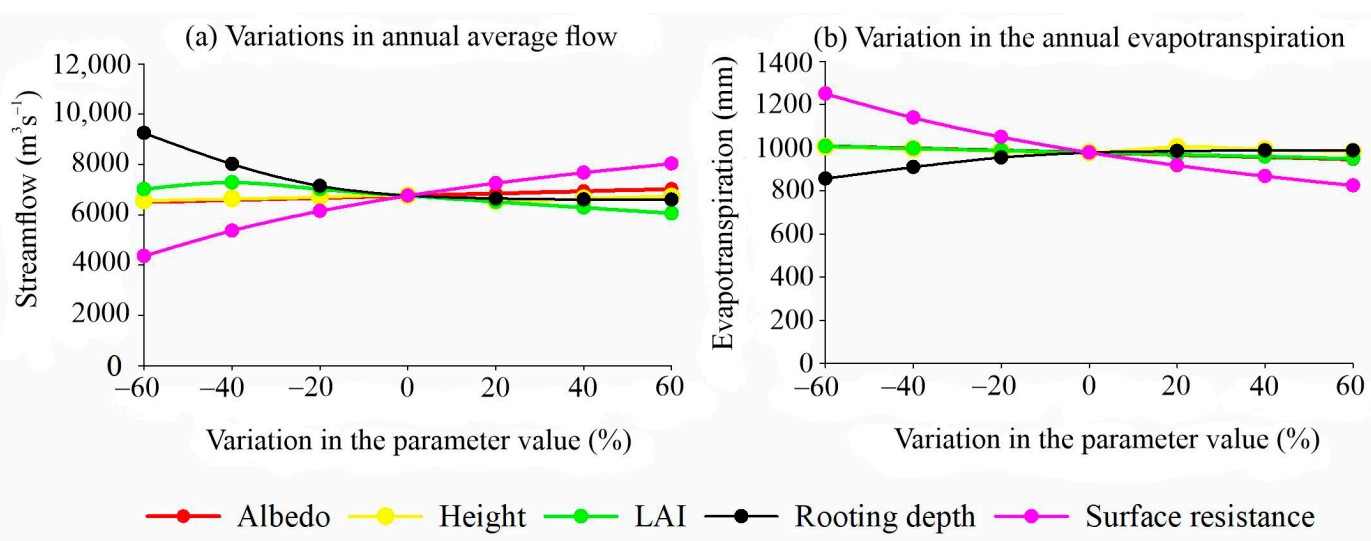

**Figure 5.** Variations in (**a**) mean annual streamflow; and (**b**) actual annual evapotranspiration resulting from the sensitivity analysis of LASH regarding the vegetation-related parameters for XRB.

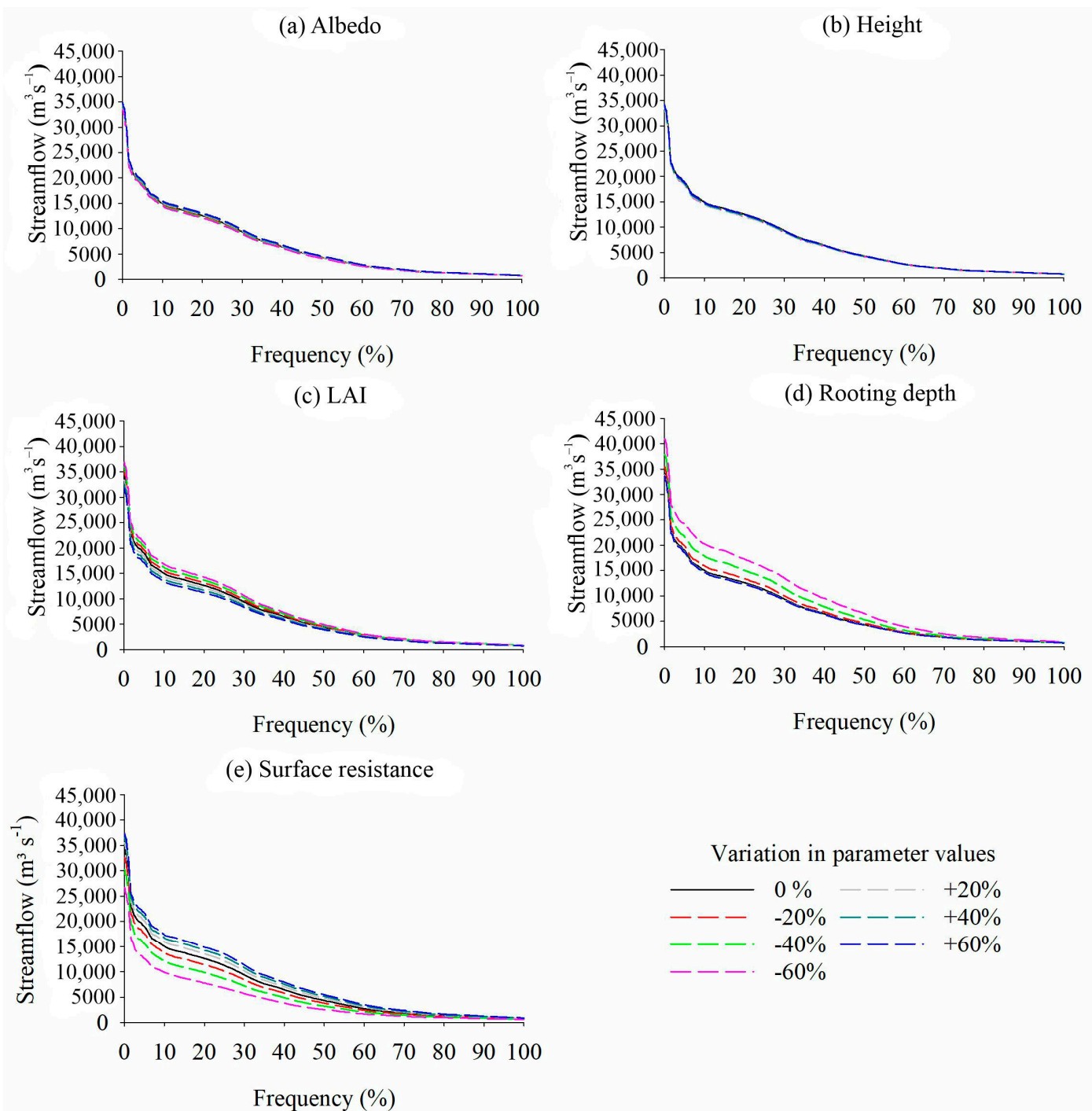

**Figure 6.** Variations in the FDC derived from the hydrographs estimated by the LASH model for the XRB considering alterations in the vegetation-related parameters.

Changes in the LAI had a more significant impact when compared to changes in the albedo and vegetation height. A reduced LAI caused increases in the mean annual streamflow of up to 13.6%, while increases provoked decreases in the mean yearly streamflow of up to −11.9%. In Figure 6c, the estimated FDC presented under- and overestimations of the maximum and intermediate streamflows (frequencies below 70%). This behavior was similar to [13] when performing a sensitivity analysis of the vegetation attributes in the LASH model. These researchers found that the LAI significantly influenced the estimated streamflows, especially the maximum streamflows.

Reductions in the Pr significantly impacted the mean annual streamflows. In the LASH model, the mechanism that explains the role of Pr in hydrological processes is linked to the water content available for evapotranspiration, as the root system controls the water balance if it is shallower than the soil depth. Therefore, Pr variations impact the water availability at the source (soil) ("source restriction"). In the estimated FDC (Figure 6d), an overestimation of the streamflows as the Pr decreased can be observed. This behavior was more evident in higher streamflows than lower streamflows. A reduction in the Pr led to an increase of up to 40% in the average annual streamflow, whereas its increase did not result in significant impacts. A Pr reduction impacted the annual evapotranspiration, decreasing it up to −15.7%.

Variations in the stomatal resistance significantly affected the average annual streamflow and annual evapotranspiration. A decrease caused significant impacts, reducing the mean annual streamflow (−35.3%) and increasing the ET (25.8%). The stomatal resistance increased the streamflow by 19.1% and decreased the ET by 15.2%. A pronounced discrepancy in the maximum streamflows was observed in the estimated FDC (Figure 6e), which agrees with [13], who concluded that this parameter causes noticeable effects on peak streamflows.

### 3.3. Hydrological Simulation of the Agribusiness Land-Use Scenarios in the XRB

The results obtained from the calibration and validation indicated that the LASH model could be used to perform hydrological simulations considering land-use changes in the XRB. The simulated hydrographs for scenarios $S_1$ and $S_2$ compared to the estimated hydrograph for $S_0$ can be observed in Figure 7, whereas the simulated hydrographs for scenarios $S_3$ and $S_4$ compared to the hydrograph for $S_0$ are in Figure 8. One can observe that, in general, land-use changes in the XRB generated an increase in streamflows, being more evident in the peak values. Basically, infiltration and soil–water storage were reduced when native forests were removed, increasing the peak streamflows and flood vulnerability [4,13].

Table 3 presents the average annual values of the water balance components from 1996 to 2005, considering the entire drainage area of the XRB. In general, the agribusiness scenarios in the XRB resulted in reductions in the ETr, It, At, and $D_{base}$ (baseflow) and increases in the $D_{sup}$ (direct surface runoff). This conclusion aligns with the findings of [38], who reported that deforestation promotes a reduction in the It and increases the direct surface runoff (overland flow) and peak flows, thus reducing the canopy evaporation from forest basins.

**Table 3.** Mean annual values of the main components of the water balance represented in the LASH model in the XRB for $S_0$ and for the agribusiness scenarios ($S_1$, $S_2$, $S_3$, and $S_4$) and respective percentage changes in relation to $S_0$.

| mm Year$^{-1}$ | $S_0$ | $S_1$ | $S_2$ | $S_3$ | $S_4$ |
|---|---|---|---|---|---|
| Rainfall | 1826.31 | 1826.31 | 1826.31 | 1826.31 | 1826.31 |
| ETr | 978.00 | 989.00 (+1.1%) | 965.07 (−1.3%) | 975.55 (−0.3%) | 939.20 (−4.0%) |
| It | 196.49 | 187.36 (−4.6%) | 173.45 (−11.7%) | 186.75 (−5.0%) | 174.61 (−11.1%) |
| At | 159.70 | 147.58 (−7.6%) | 125.49 (−21.4%) | 147.52 (−7.6%) | 132.96 (−16.7%) |
| $D_{sup}$ | 23.38 | 24.40 (+4.4%) | 30.02 (+28.4%) | 25.09 (+7.3%) | 30.34 (+29.8%) |
| $D_{base}$ | 16.37 | 16.11 (−1.6%) | 15.57 (−4.9%) | 16.23 (−0.8%) | 16.05 (−1.9%) |

ETr: actual evapotranspiration; It: interception; At: soil–water storage; $D_{sup}$: direct surface runoff (overland flow); and $D_{base}$: baseflow.

Corroborating with the results found in this study, [39] found that the evapotranspiration values can be approximately 40% lower in pasture and soybean than those obtained in native forests. However, there was an increase in the ETr in $S_1$ compared to the $S_0$ scenario. This increase was linked to the vegetation parameters adopted in the hydrological modeling, especially the RE (stomatal resistance), which had a greater sensitivity in

modeling the ETr (Figure 5). In addition, the soils of the sub-basins where the $S_1$ and $S_2$ scenarios were designed are predominantly shallow and have a reduced water storage capacity, constraining the forest ET. Therefore, the RE parameter was the main element that controlled the ET in these sub-basins.

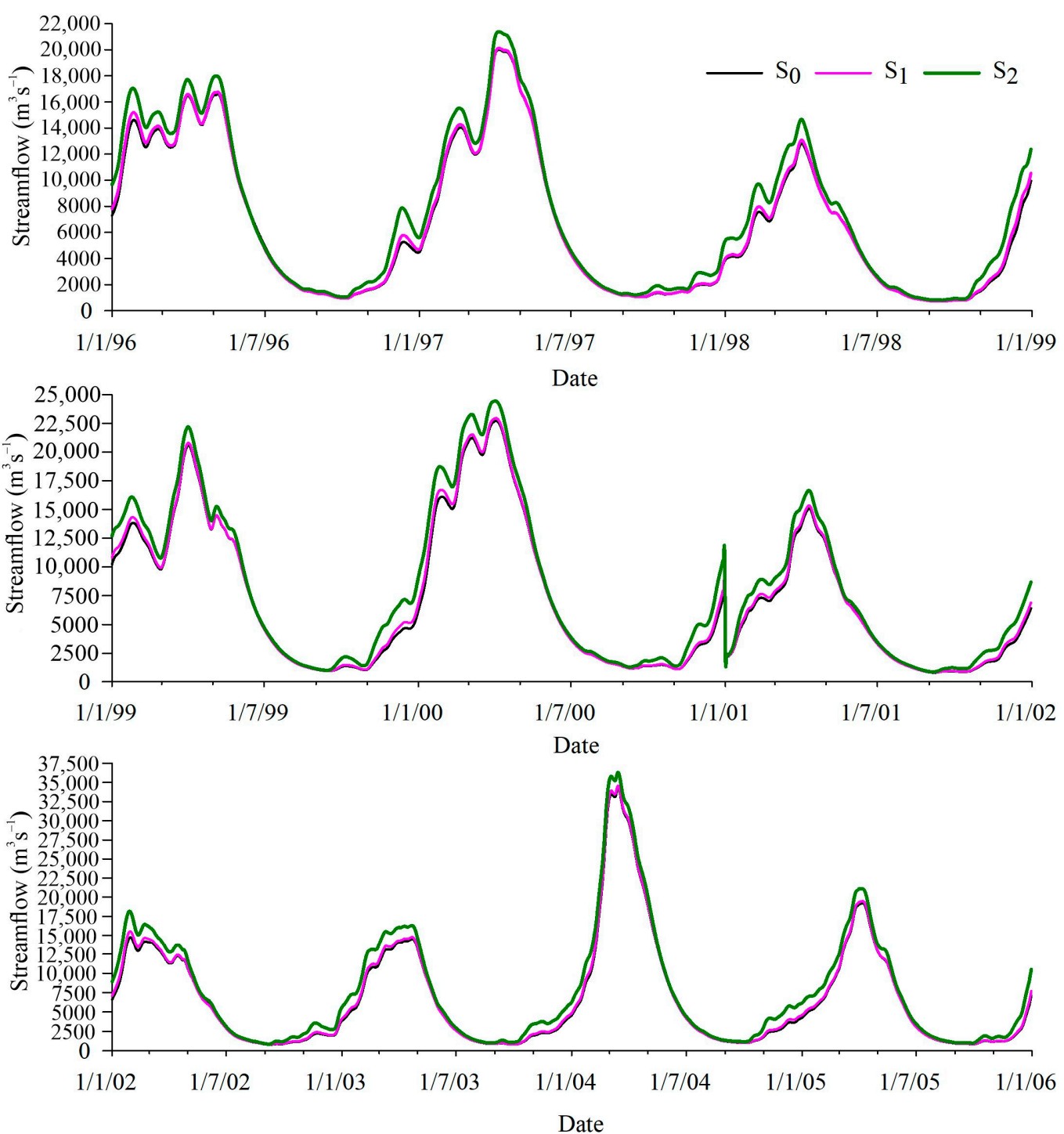

**Figure 7.** Comparison of the hydrographs estimated by the LASH model for $S_0$ and hydrographs simulated by the model for $S_1$ and $S_2$ scenarios.

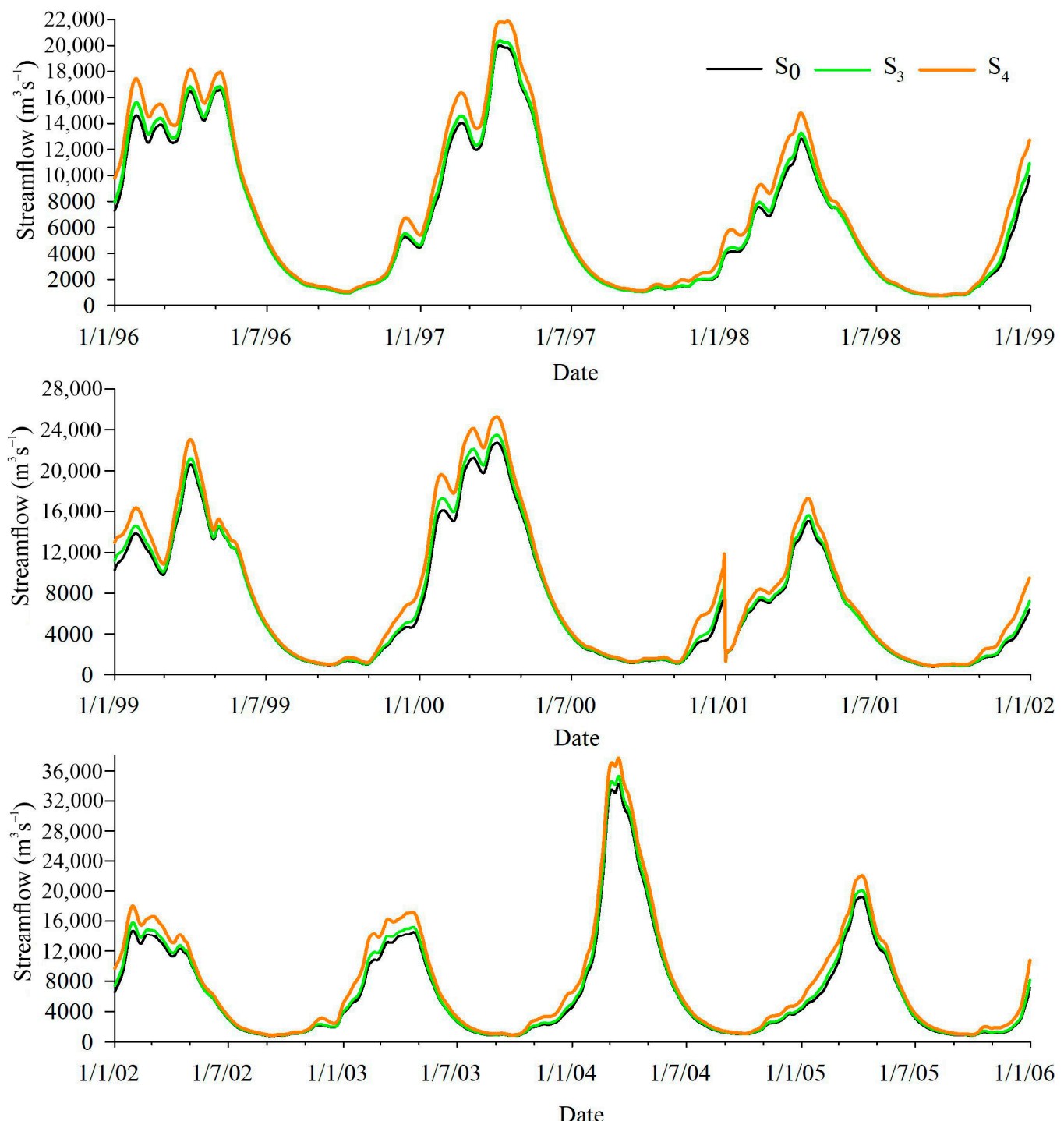

**Figure 8.** Comparison of the hydrographs estimated by the LASH model for $S_0$ and hydrographs simulated $S_3$ and $S_4$ scenarios.

Grasslands and soybean in the XRB have the dynamics of planting during the wet season (November–April) and harvesting in the dry season (May–October). In the latter, the soils covered by grasslands and soybean tend to have hydrological attributes similar to bare soils. In this period, a decreased evapotranspiration is expected due to an increase in the RE and a decrease in the LAI. This behavior was observed in the hydrological simulations for the scenarios in the XRB. In the Mortes River watershed, [7] obtained similar behavior in their simulations. For the dry season, these authors affirmed that the ETr was lower in the soybean cultivation scenarios due to LAI variation throughout the year.

Regarding the soil–water storage (At), such values indicated reductions in all the scenarios, particularly for $S_2$ (−21.4%). Besides the decrease in the Pr, another critical factor in this scenario was the soil classes where changes from forest to grassland were projected. The areas of change in $S_2$ were mostly regions with Argisols and Neosols, which are characterized by having medium and low depths, respectively, and a lower saturation and permanent wilting moisture content than Oxisols. On the other hand, the scenarios of change from forest to soybean ($S_3$ and $S_4$) predominantly consisted of Oxisols (Latosols). These soils are clayey and deeper, experience higher saturation and permanent wilting soil moisture values, and are characterized by a greater soil–water storage capacity.

All the land-use change scenarios had a decrease in baseflow. Removing native forests generated a rapid response in the direct surface runoff due to a reduction in interception and infiltration, which led to a reduction in the baseflow [30]. There is evidence that, under native forests, an improvement can be observed in soil structuring and the formation of preferential flows via biological activity and organic matter for the aggregation of soil particles [8,30].

On average, the direct surface runoff (overland flow) increased by 4.4% for S1 and 28.4% for S2 (grassland instead of native forest). The scenarios of land-use changes from native forest to soybean ($S_3$ and $S_4$) were the ones that resulted in higher increases in the direct surface runoff, corresponding to 7.3% and 30.3%, respectively, for $S_3$ and $S_4$. This fact was associated with a decrease in the rainfall interception with a consequent increase in the ET, increasing the rainfall that hit the ground, thereby generating more direct surface runoff. Thus, there were negative impacts, such as an increase in the frequency of floods and sediment transport and a reduction in the water yield in the basin and its sub-basins. The results found in the present study for the direct surface runoff in the deforestation scenarios corroborate those reported in other studies, such as [40], who found an 8% increase in the direct surface runoff in the XRB with increasing deforestation, and [41], who obtained 10–12% of the increase in the direct surface runoff in the XRB due to converting 40% of its area occupied with native forest into agriculture.

Figure 9 presents the FDC for each agribusiness scenario simulated for sub-basins 44 and 80, while Table 4 presents the respective average annual values of the water balance components from 1996 to 2005.

**Table 4.** Mean annual values of the main components of the water balance represented in the LASH model in the sub-basin 44 and sub-basin 80 for $S_0$ and the agribusiness scenarios ($S_1$, $S_2$, $S_3$, and $S_4$) and respective percentage changes concerning $S_0$.

| mm Year$^{-1}$ | Sub-Basin 44 | | | Sub-Basin 80 | | |
|---|---|---|---|---|---|---|
| | $S_0$ | $S_1$ | $S_2$ | $S_0$ | $S_3$ | $S_4$ |
| Rainfall | 2158.1 | 2158.1 | 2158.1 | 1678.3 | 1678.3 | 1678.3 |
| ETr | 1000.0 | 1063.5 (+6.3%) | 1027.7 (+2.8%) | 1212.2 | 1187.7 (−2.0%) | 957.9 (−21.0%) |
| It | 240.0 | 195.6 (−18.5%) | 136.1 (−43.3%) | 174.9 | 142.2 (−18.7%) | 97.2 (−44.4%) |
| At | 195.6 | 132.0 (−32.5%) | 49.1 (−74.9%) | 91.2 | 84.6 (−7.2%) | 40.2 (−55.9%) |
| $D_{sup}$ | 405.1 | 469.5 (+15.9%) | 708.9 (+75.0%) | 99.6 | 141.6 (+42.3%) | 410.5 (+312.3%) |
| $D_{base}$ | 233.2 | 219.6 (−5.8%) | 195.2 (−16.3%) | 128.2 | 134.0 (+4.5%) | 144.2 (+12.5%) |

ETr: actual evapotranspiration; It: interception; At: soil-water storage; $D_{sup}$: direct surface runoff (overland flow); and $D_{base}$: baseflow.

For the simulations of sub-basin 44, it is possible to observe that the scenarios of increasing grasslands caused increases in the peak streamflows. This behavior for the Tapajós River Basin was also found by [6], reporting that increases in grassland areas cause an increase in the streamflow magnitudes, except for very low streamflows (>95% exceedance in the FDC). Likewise, when analyzing the conversion of native forests into grasslands in the Mortes River watershed, [7] the observed streamflow increased.

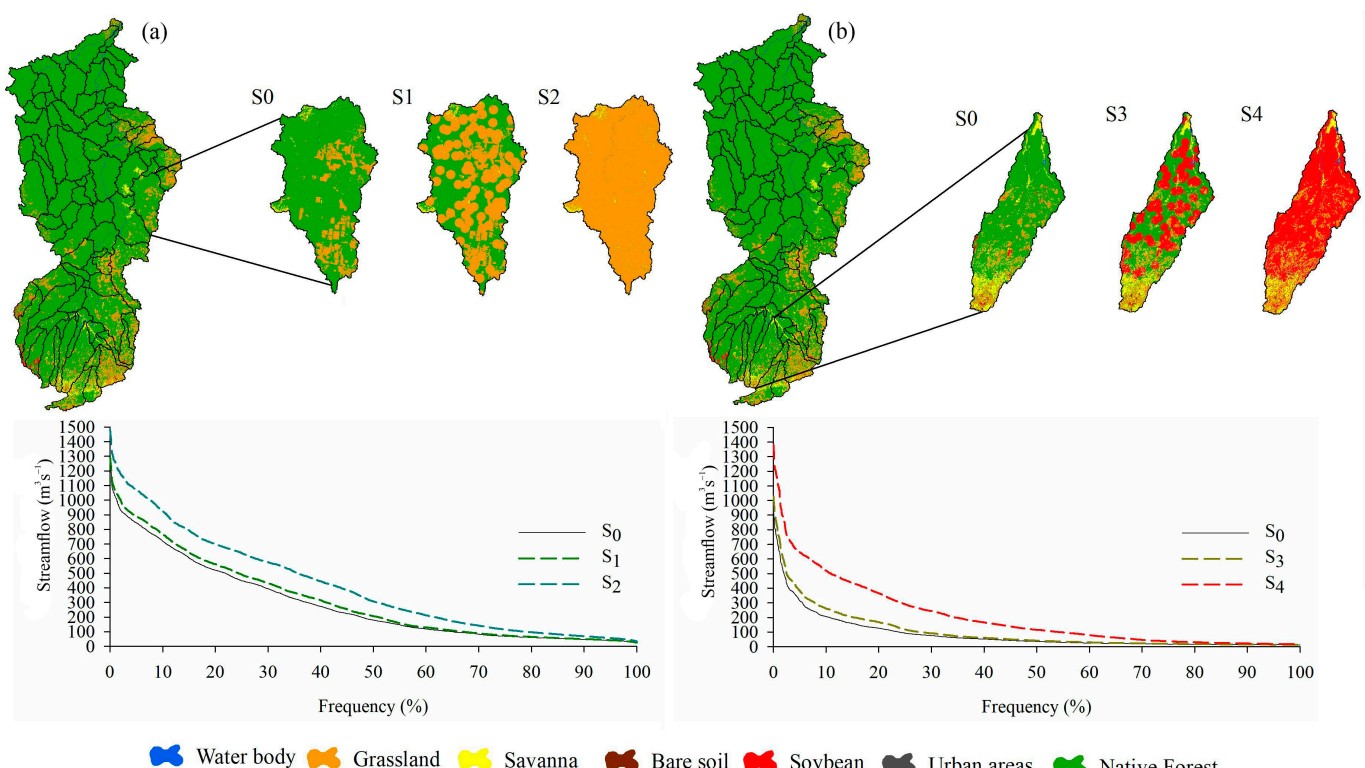

**Figure 9.** Hydrographs simulated by LASH at the sub-basins 44 (**a**) and 80 (**b**) considering $S_1$ and $S_2$ compared to the hydrograph estimated for $S_0$ in the same sub-basins.

The increasing soybean area scenario generally provoked increased streamflows at sub-basin 80. The headwater sub-basins of the XRB were assessed by [39]. They observed that basins occupied with soybean cultivation had an average annual observed streamflow three times higher than that of basins with native forests, mainly because of a reduction in evapotranspiration. Increasing the soybean areas in the Mortes River watershed tended to significantly experience an increase in the maximum and average streamflows and a decrease in the minimum ones, while the opposite behavior was identified for the forest areas in the watershed [7]. This behavior occurred because of the lower LAI and Pr values, the higher albedo values, reducing the evapotranspiration and increasing the streamflows in watersheds with a predominance of soybean areas [7,39].

### 3.4. Limitations of Hydrological Models in Simulating the Land-Use Impacts at the Basin-Scale

Most hydrological models used to simulate the impacts of land-use changes on a basin's hydrology have limitations. The following limitations connected with hydrological models should be mentioned [4]: (i) they usually take into account only a few vegetation-related parameters, and (ii) the models are unable to describe how land-use changes impact the dynamics of the soil–water infiltration and, consequently, the groundwater recharge.

Some models used for the hydrological simulation of land-use scenarios might not understand the characteristics of tropical climate forests in their databases, requiring some adaptations [5]. Hydrological models are associated with adequately representing more complex soil classes, especially highly parameterized models [6].

Furthermore, hydrological models with different spatial (local, mesoscale, or global) or temporal scales (daily, seasonal, annual, or multi-year) could provide different results, especially when simulating the ET [42]. The sensitivity of these hydrological impact estimations from land-use changes may be influenced by the size of the basin, because changes in the sub-basins located far from the basin's outlet may not cause significant changes in the total runoff [30].

Evaluations of the role of forests in the water balance have been widely discussed in the literature, principally considering the perspective of water demand for consumption, energy generation, and irrigation. However, analyses considering this perspective can erroneously indicate that forests reduce the water available for downstream activities compared to other land uses, such as grasslands and soybean. In the deforestation scenarios projected in the present study, it was possible to observe a reduction in the baseflow, meaning a reduction in the infiltration and recharge processes, i.e., the simulations demonstrated that there would be a reduction in the water availability in the basin, especially in those areas more affected by the projected changes, and a potential increase in flooded areas, erosion, and sediment transport.

In this sense, it is relevant to note that forests are one of the main drivers of the hydrological cycle at different spatial scales. Furthermore, they are essential in maintaining the climate, as water vapor supplies the atmosphere and influences the rainfall regime. Thus, current deforestation rates can significantly alter the magnitude of the hydrological cycle components in the XRB and, by extension, the Amazon Rainforest and Brazilian Cerrado.

## 4. Conclusions

a.  The LASH model showed a good performance in the XRB (NS > 0.85 and NSlog > 0.86 in both calibration and validation), including the Proxy Basin test (NS = 0.77; NSlog = 0.88), which allows for understanding the hydrological processes' simulation in this Amazon basin.

b.  Regarding the hydrological analysis of the agribusiness land-use scenarios, land use based on deforestation in the XRB would increase the direct surface runoff (from 4.4%–S1 to 29.8%–S4). A reduction in the baseflow was mainly observed in the grassland scenarios, being −1.6% and −4.9% for S1 and S2, respectively, which was clearer for the sub-basins in the headwater region of the basin, where the scenarios were more effective.

c.  The peak flows were more pronounced for the S2 and S4 scenarios, which considers 100% of the deforestation for grasslands and soybean, respectively, and where agribusiness activities have been more frequent.

d.  The baseflow could be significantly reduced in all the projected scenarios, especially for S2 in the headwater sub-basins (−16.3%), which can compromise the water yield in the basin.

**Author Contributions:** Conceptualization, C.R.M. and Z.A.C.; methodology, Z.A.C., C.R.M. and S.B.; software, M.M.V., M.M.M. and S.B.; validation, C.R.M. and J.A.G.; formal analysis, C.R.M., Z.A.C. and S.B.; resources, Z.A.C. and S.B.; data curation, C.R.M.; writing—original draft preparation, C.R.M., J.A.G. and S.B.; visualization, Z.A.C.; supervision, C.R.M.; project administration, C.R.M.; funding acquisition, C.R.M. All authors have read and agreed to the published version of the manuscript.

**Funding:** This research was funded by Conselho Nacional de Pesquisa e Desenvolvimento Científico—CNPq, grant number 401156/2022-2 and The APC was funded by Carlos R. Mello.

**Data Availability Statement:** The data presented in this study are available on request from the corresponding author.

**Conflicts of Interest:** The authors declare no conflict of interest. The funders had no role in the design of the study; in the collection, analyses, or interpretation of data; in the writing of the manuscript; or in the decision to publish the results.

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
