# Peer review of "A Modeling Approach for Analyzing the Hydrological Impacts of the Agribusiness Land-Use Scenarios in an Amazon Basin"

_land, doi:10.3390/land12071422_

Round 1

Reviewer 1 Report

The Brazilian amazon forest has a pivotal role in the global water cycle. Deforestation has a significant impact on the water cycle. In this paper, authors simulate the hydrological processes in the Brazilian XRB based on the LASH model and set up four agribusiness scenarios to explore the hydrological effects of deforestation. The study is relatively clear, the data are detailed, and the method is more accurate. However, the following problems still exist, and it is recommended to consider revisions.

1. The abstract needs to be rewritten. It is suggested to add some quantitative description of the results. Reflect the specific impacts of land use/cover change.

2. L243-250, and also Figure 3, how is the location of land use/cover type change determined in these 4 scenarios?

3. Although the NS statistics are high, the runoff curve (Fig. 4a) shows relatively poor capture for flood peaks. If possible, it can be optimized again appropriately. Because the change of land use/cover type still has a great influence on the flood peak.

4. L311, what is the case of Citation 36?

5. L334-336, the LASH model performs "very well" in the IRB, which is your own research result. Why do you need to add reference 34?

6. Sec.3.2, why did you choose 5 vegetation parameters such as Albedo to do the sensitivity analysis? Are these 5 parameters the more important parameters in the model?

7. Figure 7, the differences between the simulation results and the observation results of each scenario are not obvious, can we add subplots for enlarging the local differences to show? This can highlight the impact of land use/change on runoff.

8. Table 3, the full name of ETr and other abbreviations, it is recommended to fill in appropriately.

9. Table 4, why does the conversion of forest land to grassland lead to an increase in ETr, while the conversion of forest land to soybean leads to a decrease in ETr?

no

Reviewer 2 Report

Dear Authors,

The manuscripted entitled "A modeling approach for analyzing the hydrological impacts of the agribusiness land use scenarios in an Amazon basin" fits in the purposes of the special issue.

Please find my review as comments in the text.

Reviewer 3 Report

Manuscript title:   A modeling approach for analyzing the hydrological impacts of the agribusiness land use scenarios in an Amazon basin

Manuscript id: land-2482891

Authors:  Cunha et al.

The manuscript is particularly strong regarding the less studied topic and the experimental setup on hydrological impacts of modeling ……. The manuscript regarding the topic and results presented is of interest to environmental science community and revisions based on the comments below are recommended before considering for publication.

Major comments

·       The unit/abbreviation is not mentioned before, consider defining the abbreviation when mentioned for the first time…. Please check throughout the manuscript to define the abbreviations.

·       Line 80-86, the aim or hypothesis of the study is clear, however, the approach is missing ….

·       Lake of scientific literature to support the statements and findings throughout the manuscript…... I have made some suggestions for that and more need it….

·       More information is needed for ALL TABLE captions and define the abbreviation and units that are used. And adjust the significant figures for the table and manuscript.

·     ·       I have a major concern about the results and discussion section. The authors describe the results and compare the results with previous studies, however, insight mechanisms are still insufficient.

·       This section is repeating information already presented and explaining things in an unnecessarily complicated way. The quality of the manuscript would benefit from the whole section being condensed.

Detailed comments:

Abstract

If the unit/abbreviation is not mentioned before, consider defining the abbreviation when mentioned for the first time.

Introduction:

Line 30-36: A reference is needed here, for example, you can use: https://doi.org/10.1016/j.cropro.2013.10.022

https://doi.org/10.3390/land6030053

Line 48: A reference is needed here, for example, you can use: https://doi.org/10.1016/j.scitotenv.2022.155283

Line 63-72: A reference is needed here.

Line 64-69: A complicated sentence, please revise and check the grammar

In MM section

Literature references are missing for all sub-section. It would be better to cite the references that the procedure adopted.

Additional info is needed for the table caption, most importantly significant figures.

In MM section, what is the quality control (QC) data? There is no mention of the QC.

What is the accuracy of the instruments, recovery, LOD, and LOQ ……. These parameters are needed to report the efficiency of any analytical system.

In general, how many times you’ve recorded the data,? duplicate? Triplicate?..... what you mentioned in the text is not clear, please elaborate more on this

R&D section

These sections are repeating information already presented and explain things in an unnecessarily complicated way. The quality of the manuscript would benefit from the whole section being condensed, Line 309-312, Line 371-382, Line 391-411, Line 450-492, Line 550-562……

Conclusion

The section should not be a summary of your study or an extension of the discussion. This section should illustrate the mechanistic links of findings of this study. The conclusions should answer the hypothesis of your study and should focus on the implication of your findings. Remember that the conclusions must be self-explanatory. This section should still highlight the novelty and implication of your study also.

 Grammar and punctuation issuers need to be addressed. I have selected/mentioned some as examples.

Round 2

Reviewer 1 Report

According to my comments, the authors made all the changes. I agree that this manuscript is published in Land.

Reviewer 3 Report

The revised manuscript has improved compared to the original version. The authors tried to address my questions as much as possible. I recommend the manuscript to be published!